# Sample-efficient Multi-objective Molecular Optimization with GFlowNets

**Yiheng Zhu**[1]*, **Jialu Wu**[2]*, **Chaowen Hu**[3], **Jiahuan Yan**[1],
**Chang-Yu Hsieh**[2]†, **Tingjun Hou**[2]†, **Jian Wu**[4,5,6]†

[1]College of Computer Science and Technology, Zhejiang University
[2]College of Pharmaceutical Sciences, Zhejiang University
[3]Polytechnic Institute, Zhejiang University
[4]Second Affiliated Hospital School of Medicine, Zhejiang University
[5]School of Public Health, Zhejiang University
[6]Institute of Wenzhou, Zhejiang University
{zhuyiheng2020, jialuwu, ChaowenHu, jyansir, kimhsieh, tingjunhou, wujian2000}@zju.edu.cn

## Abstract

Many crucial scientific problems involve designing novel molecules with desired properties, which can be formulated as a black-box optimization problem over the *discrete* chemical space. In practice, multiple conflicting objectives and costly evaluations (e.g., wet-lab experiments) make the *diversity* of candidates paramount. Computational methods have achieved initial success but still struggle with considering diversity in both objective and search space. To fill this gap, we propose a multi-objective Bayesian optimization (MOBO) algorithm leveraging the hypernetwork-based GFlowNets (HN-GFN) as an acquisition function optimizer, with the purpose of sampling a diverse batch of candidate molecular graphs from an approximate Pareto front. Using a single preference-conditioned hypernetwork, HN-GFN learns to explore various trade-offs between objectives. We further propose a hindsight-like off-policy strategy to share high-performing molecules among different preferences in order to speed up learning for HN-GFN. We empirically illustrate that HN-GFN has adequate capacity to generalize over preferences. Moreover, experiments in various real-world MOBO settings demonstrate that our framework predominantly outperforms existing methods in terms of candidate quality and sample efficiency. The code is available at `https://github.com/violet-sto/HN-GFN`.

## 1 Introduction

Designing novel molecular structures with desired properties, also referred to as molecular optimization, is a crucial task with great application potential in scientific fields ranging from drug discovery to material design. Molecular optimization can be naturally formulated as a black-box optimization problem over the *discrete* chemical space, which is combinatorially large [52]. Recent years have witnessed the trend to leverage computational methods, such as deep generative models [31] and combinatorial optimization algorithms [62, 30], to facilitate optimization. However, the applicability of most prior approaches in real-world scenarios is hindered by two practical constraints: (i) Chemists commonly seek to optimize multiple properties of interest simultaneously [60, 33]. For example, in addition to effectively inhibiting a disease-associated target, an ideal drug is desired to be easily synthesizable and non-toxic. Unfortunately, as objectives often conflict, in most cases, there is no single

---

*Equal contribution.
†Corresponding authors.

37th Conference on Neural Information Processing Systems (NeurIPS 2023).

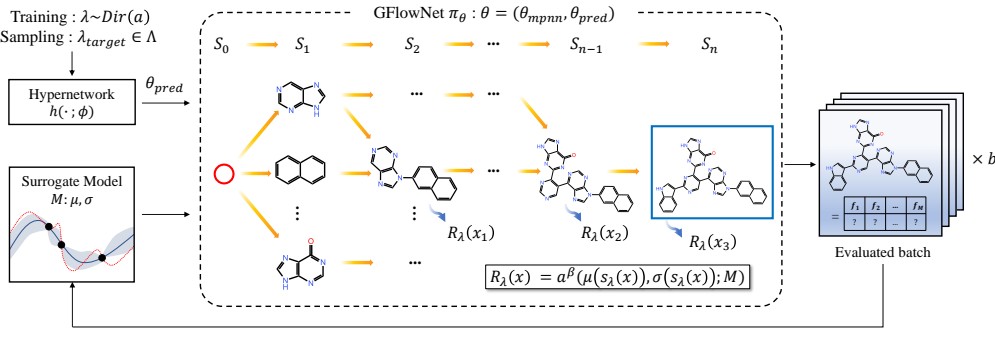

Training: $\lambda \sim Dir(\alpha)$
Sampling: $\lambda_{target} \in \Lambda$

GFlowNet $\pi_\theta : \theta = (\theta_{mpnn}, \theta_{pred})$

$S_0 \quad S_1 \quad S_2 \quad \cdots \quad S_{n-1} \quad S_n$

Hypernetwork $h(\cdot\,;\phi)$

$\theta_{pred}$

Surrogate Model $M : \mu, \sigma$

$R_\lambda(x_1) \qquad R_\lambda(x_2) \qquad R_\lambda(x_3)$

$R_\lambda(x) = a^\beta(\mu(s_\lambda(x)), \sigma(s_\lambda(x)); M)$

$\times b$

Evaluated batch

$\bigcirc$ *represents the initial state and* $\square$ *represents the complete object.*

Figure 1: MOBO loop for molecular optimization using a surrogate model $\mathcal{M}$ for uncertainty estimation and HN-GFN for acquisition function optimization. In each round, the policy $\pi_\theta$ is trained with reward function $R_\lambda$, where $\lambda$ is sampled from $\mathrm{Dir}(\alpha)$ per iteration. A new batch of candidates is sampled from the approximate Pareto front according to target preference vectors $\lambda_{\text{target}} \in \Lambda$.

optimal solution, but rather a set of Pareto optimal solutions defined with various trade-offs [18, 45]. (ii) Realistic oracles (e.g., wet-lab experiments and high-fidelity simulations) require substantial costs to synthesize and evaluate molecules [22]. Hence, the number of oracle evaluations is notoriously limited. In such scenarios, the diversity of candidates is a critical consideration.

Bayesian optimization (BO) [34, 54] provides a sample-efficient framework for globally optimizing expensive black-box functions. The core idea is to construct a cheap-to-evaluate *surrogate model* approximating the true objective function (also known as the *oracle*) on the observed dataset and optimize an *acquisition function* (built upon the surrogate model) to obtain informative candidates with high utility for the next round of evaluations. Due to the common large-batch and low-round settings in biochemical experiments [3], batch BO is prioritized to shorten the entire cycle of optimization [25]. MOBO also received broad attention and achieved promising performance in continuous problems by effectively optimizing differentiable acquisition functions [12, 13]. Nevertheless, it is less prominent in discrete problems, where no gradients can be leveraged to navigate the search space.

Although existing discrete molecular optimization methods can be adopted as the *acquisition function optimizer* to alleviate this issue, they can hardly consider diversity in both search and objective space: 1) many neglect the diversity of proposed candidates in search space [30, 33]; 2) many multi-objective methods simply rely on a predefined scalarization function (e.g., mean) to turn the multi-objective optimization (MOO) problem into a single-objective one [61, 20]. As the surrogate model cannot exactly reproduce the oracle's full behaviors and the optimal trade-off is unclear before optimization (even with domain knowledge), it is desired to not only propose candidates that bring additional information about the search space but also explore more potential trade-offs of interest. To achieve this goal, we explore how to extend the recently proposed generative flow networks (GFlowNets) [6, 7], a class of generative models that aim to learn a stochastic policy for sequentially constructing objects with a probability proportional to a reward function, to facilitate multi-objective molecular optimization. Compared with traditional combinatorial optimization methods, GFlowNets possess merit in generating diverse and high-reward objects, which has been verified in single-objective problems [6, 28].

In this work, we present a MOBO algorithm based on GFlowNets for sample-efficient multi-objective molecular optimization. We propose a hypernetwork-based GFlowNet (HN-GFN) as the acquisition function optimizer within MOBO to sample a diverse batch of candidates from an approximate Pareto front. Instead of defining a fixed reward function as usual in past work [6], we train a unified GFlowNet on the distribution of reward functions (*random scalarizations* parameterized by *preference vectors*) and control the policy using a single preference-conditioned hypernetwork. While sampling candidates, HN-GFN explores various trade-offs between competing objectives flexibly by varying the input preference vector. Inspired by hindsight experience replay [2], we further introduce a hindsight-like off-policy strategy to share high-performing molecules among different preferences

and speed up learning for HN-GFN. As detailed in our reported experiments, we first empirically verify that HN-GFN is capable of generalizing over preference vectors, then apply the proposed framework to real-world scenarios and demonstrate its effectiveness and efficiency over existing methods. Our key contributions are summarized below:

- We introduce a GFlowNet-based MOBO algorithm to facilitate real-world molecular optimization. We propose HN-GFN, a conditional variant of GFlowNet that can efficiently sample candidates from an approximate Pareto front.

- We introduce a hindsight-like off-policy strategy to speed up learning in HN-GFN.

- Experiments verify that our MOBO algorithm based on HN-GFN can find a high-quality Pareto front more efficiently compared to state-of-the-art baselines.

## 2   Related Work

**Molecular optimization.**   Molecular optimization has been approached with a wide variety of computational methods, which can be mainly grouped into three categories: 1) **Latent space optimization (LSO)** methods [24, 31, 58, 66] perform the optimization over the low-dimensional continuous latent space learned by generative models such as variational autoencoders (VAEs) [36]. These methods require the latent representations to be discriminative, but the training of the generative model is decoupled from the optimization objectives, imposing challenges for optimization [58]. Instead of navigating the latent space, combinatorial optimization methods search for the desired molecular structures directly in the explicit discrete space with 2) **evolutionary algorithms** [30] and 3) **reinforcement learning (RL)** [62] guiding the search. However, most prior methods only focus on a single property, from non-biological properties such as drug-likeliness (QED) [8] and synthetic accessibility (SA) [19], to biological properties that measure the binding energy to a protein target [6]. Multi-objective molecular optimization has recently received wide attention [33, 61, 20]. For example, MARS [61] employs Markov chain Monte Carlo (MCMC) sampling to find novel molecules satisfying several properties. However, most approaches require a notoriously large number of oracle calls to evaluate molecules on the fly [33, 61]. In contrast, we tackle this problem in a sample-efficient manner.

**GFlowNet.**   GFlowNets [6, 7] aim to sample composite objects proportionally to a reward function, instead of maximizing it as usual in RL [56]. GFlowNets are related to the MCMC methods due to the same objective, while amortizing the high cost of sampling (mixing between modes) over training a generative model. GFlowNets have made impressive progress in various applications, such as biological sequence design [28], discrete probabilistic modeling [63], and Bayesian structure learning [15]. While the concept of conditional GFlowNet was originally discussed in Bengio et al. [7], we are the first to study and instantiate this concept for MOO, in parallel with Jain et al. [29]. Compared with them, we delicately design the conditioning mechanism and propose a hindsight-like off-policy strategy that is rarely studied for MOO (in both the RL and GFlowNet literature). Appendix D provides a detailed comparison.

**Bayesian optimization for discrete spaces.**   The application of BO in discrete domains has proliferated in recent years. It is much more challenging to construct surrogate models and optimize acquisition functions in discrete spaces, compared to continuous spaces. One common approach is to convert discrete space into continuous space with generative models [24, 16, 44]. Besides, one can directly define a Gaussian Process (GP) with discrete kernels [46, 53] and solve the acquisition function optimization problem with evolutionary algorithms [35, 57].

**Multi-objective Bayesian optimization.**   BO has been widely used in MOO problems for efficiently optimizing multiple competing expensive black-box functions. Most popular approaches are based on hypervolume improvement [12], random scalarizations [37, 49], and entropy search [27, 5]. While there have been several approaches that take into account parallel evaluations [11, 38] and diversity [38], they are limited to continuous domains.

## 3 Background

### 3.1 Problem formulation

We address the problem of searching over a discrete chemical space $\mathcal{X}$ to find molecular graphs $x \in \mathcal{X}$ that maximize a *vector-valued objective* $f(x) = \big(f_1(x), f_2(x), \ldots, f_M(x)\big) : \mathcal{X} \to \mathbb{R}^M$, where $f_m$ is a black-box function (also known as the oracle) evaluating a certain property of molecules. Practically, realistic oracles are extremely expensive to evaluate with either high-fidelity simulations or wet-lab experiments. We thus propose to perform optimization in as few oracle evaluations as possible, since the sample efficiency is paramount in such a scenario.

There is typically no single optimal solution to the MOO problem, as different objectives may contradict each other. Consequently, the goal is to recover the *Pareto front* – the set of *Pareto optimal* solutions which cannot be improved in any one objective without deteriorating another [18, 45]. In the context of maximization, a solution $f(x)$ is said to *Pareto dominates* another solution $f(x')$ iff $f_m(x) \geq f_m(x') \; \forall m = 1, \ldots, M$ and $\exists m'$ such that $f_{m'}(x) > f_{m'}(x')$, and we denote $f(x) \succ f(x')$. A solution $f(x^*)$ is *Pareto optimal* if not Pareto dominated by any solution. The Pareto front can be written as $\mathcal{P}^* = \{f(x^*) : \{f(x) : f(x) \succ f(x^*)\} = \emptyset\}$.

The quality of a finite approximate Pareto front $\mathcal{P}$ is commonly measured by *hypervolume* [67] – the M-dim Lebesgue measure $\lambda_M$ of the space dominated by $\mathcal{P}$ and bounded from below by a given reference point $\boldsymbol{r} \in \mathbb{R}^M$: $HV(\mathcal{P}, \boldsymbol{r}) = \lambda_M(\cup_{i=1}^{|\mathcal{P}|}[\boldsymbol{r}, y_i])$, where $[\boldsymbol{r}, y_i]$ denotes the hyper-rectangle bounded by $\boldsymbol{r}$ and $y_i = f(x_i)$.

### 3.2 Batch Bayesian optimization

Bayesian optimization (BO) [54] provides a model-based iterative framework for sample-efficient black-box optimization. Given an observed dataset $\mathcal{D}$, BO relies on a Bayesian *surrogate model* $\mathcal{M}$ to estimate a posterior distribution over the true oracle evaluations. Equipped with the surrogate model, an *acquisition function* $a : \mathcal{X} \to \mathbb{R}$ is induced to assign the utility values to candidate objects for deciding which to be evaluated next on the oracle. Compared with the costly oracle, the cheap-to-evaluate acquisition function can be efficiently optimized. We consider the scenario where the oracle is given an evaluation budget of $N$ rounds with fixed batches of size $b$.

To be precise, we have access to a random initial dataset $\mathcal{D}_0 = \{(x_i^0, y_i^0)\}_{i=1}^n$, where $y_i^0 = f(x_i^0)$ is a true oracle evaluation. In each round $i \in \{1, \ldots, N\}$, the acquisition function is maximized to yield a batch of candidates $\mathcal{B}_i = \{x_j^i\}_{j=1}^b$ to be evaluated in parallel on the oracle $y_j^i = f(x_j^i)$. The observed dataset $\mathcal{D}_{i-1}$ is then augmented for the next round: $\mathcal{D}_i = \mathcal{D}_{i-1} \cup \{(x_j^i, y_j^i)\}_{j=1}^b$.

## 4 Method

In this section, we present the proposed MOBO algorithm based on hypernetwork-based GFlowNet (HN-GFN), shown in Figure 1. Due to the space limitation, we present the detailed algorithm in Appendix A. Our key idea is to extend GFlowNets as the acquisition function optimizer for MOBO, with the objective of sampling a diverse batch of candidates from the approximate Pareto front. To begin, we introduce GFlowNets in the context of molecule design, then describe how GFlowNet can be biased by a preference-conditioned hypernetwork to sample molecules according to various trade-offs between objectives. Next, we propose a hindsight-like off-policy strategy to speed up learning in HN-GFN. Finally, we introduce the surrogate model and acquisition function.

### 4.1 Preliminaries

GFlowNets [6] seek to learn a stochastic policy $\pi$ for sequentially constructing discrete objects $x \in \mathcal{X}$ with a probability $\pi(x) \propto R(x)$, where $\mathcal{X}$ is a compositional space and $R : \mathcal{X} \to \mathbb{R}_{\geq 0}$ is a non-negative reward function. The generation process of object $x \in \mathcal{X}$ can be represented by a sequence of discrete *actions* $a \in \mathcal{A}$ that incrementally modify a partially constructed object, which is denoted as *state* $s \in \mathcal{S}$. Let the generation process begin at a special initial state $s_0$ and terminate with a special action indicating that the object is complete ($s = x \in \mathcal{X}$), the construction of an object $x$ can be defined as a complete trajectory $\tau = (s_0 \to s_1 \to \cdots \to s_n = x)$.

Following fragment-based molecule design [6, 61], the molecular graphs are generated by sequentially attaching a fragment, which is chosen from a predefined vocabulary of building blocks, to an atom of the partially constructed molecules. The maximum trajectory length is 8, with the number of actions varying between 100 and 2000 depending on the state, making $|\mathcal{X}|$ up to $10^{16}$. There are multiple action sequences leading to the same state, and no fragment removal actions, the space of possible action sequences can thus be denoted by a directed acyclic graph (DAG) $\mathcal{G} = (\mathcal{S}, \mathcal{E})$, where the edges in $\mathcal{E}$ are transitions $s \to s'$ from one state to another. To learn the aforementioned desired policy, Bengio et al. [6] propose to see the DAG structure as a *flow network*.

**Markovian flows.** Bengio et al. [7] first define a *trajectory flow* $F : \mathcal{T} \to \mathbb{R}_{\geq 0}$ on the set of all complete trajectories $\mathcal{T}$ to measure the unnormalized density. The *edge flow* and *state flow* can then be defined as $F(s \to s') = \sum_{s \to s' \in \tau} F(\tau)$ and $F(s) = \sum_{s \in \tau} F(\tau)$, respectively. The trajectory flow $F$ determines a probability measure $P(\tau) = \frac{F(\tau)}{\sum_{\tau \in \mathcal{T}} F(\tau)}$. If flow $F$ is *Markovian*, the forward transition probabilities $P_F$ can be computed as $P_F(s'|s) = \frac{F(s \to s')}{F(s)}$.

**Flow matching.** A flow is *consistent* if the following *flow consistency equation* is satisfied $\forall s \in \mathcal{S}$:

$$F(s) = \sum_{s' \in Pa_{\mathcal{G}}(s)} F(s' \to s) = R(s) + \sum_{s'':s \in Pa_{\mathcal{G}}(s'')} F(s \to s'') \tag{1}$$

where $Pa_{\mathcal{G}}(s)$ is a set of parents of $s$ in $\mathcal{G}$. As proved in Bengio et al. [6], if the flow consistency equation is satisfied with $R(s) = 0$ for non-terminal state $s$ and $F(x) = R(x) \geq 0$ for terminal state $x$, a policy $\pi$ defined by the forward transition probability $\pi(s'|s) = P_F(s'|s)$ samples object $x$ with a probability $\pi(x) \propto R(x)$. GFlowNets propose to approximate the edge flow $F(s \to s')$ using a neural network $F_\theta(s, s')$ with enough capacity, such that the flow consistency equation is respected at convergence. To achieve this, Bengio et al. [6] define a temporal difference-like [56] learning objective, called flow-matching (FM):

$$\mathcal{L}_\theta(s, R) = \left( \log \frac{\sum_{s' \in Pa_{\mathcal{G}}(s)} F_\theta(s', s)}{R(s) + \sum_{s'':s \in Pa_{\mathcal{G}}(s'')} F_\theta(s, s'')} \right)^2 \tag{2}$$

One can use any exploratory policy $\widetilde{\pi}$ with full support to sample training trajectories and obtain the consistent flow $F_\theta(s, s')$ by minimizing the FM objective [6]. Consequently, a policy defined by this approximate flow $\pi_\theta(s'|s) = P_{F_\theta}(s'|s) = \frac{F_\theta(s \to s')}{F_\theta(s)}$ can also sample objects $x$ with a probability $\pi_\theta(x)$ proportionally to reward $R(x)$. Practically, the training trajectories are sampled from a mixture between the current policy $P_{F_\theta}$ and a uniform distribution over allowed actions [6]. We adopt the FM objective in this work because the alternative trajectory balance [43] was also examined but gave a worse performance in early experiments. Note that more advanced objectives such as subtrajectory balance [42] can be employed in future work.

### 4.2 Hypernetwork-based GFlowNets

Our proposed HN-GFN aims at sampling a diverse batch of candidates from the approximate Pareto front with a unified model. A common approach to MOO is to decompose it into a set of scalar optimization problems with scalarization functions and apply standard single-objective optimization methods to gradually approximate the Pareto front [37, 64]. Here we consider the weighted sum (WS): $s_\lambda(x) = \sum_i \lambda_i f^i(x)$ and Tchebycheff [45]: $s_\lambda(x) = \max_i \lambda_i f^i(x)$, where $\lambda = (\lambda_i, \cdots, \lambda_M)$ is a preference vector that defines the trade-off between the competing properties.

To support parallel evaluations, one can simultaneously obtain candidates according to different scalarizations. Practically, this approach hardly scales efficiently with the number of objectives for discrete problems. Taking GFlowNet as an example, we need to train multiple GFlowNets separately for each choice of the reward function $R_\lambda(x) = s_\lambda(x)$ to cover the objective space:

$$\theta_\lambda^* = \arg \min_\theta \mathbb{E}_{s \in \mathcal{S}} \mathcal{L}_\theta(s, R_\lambda) \tag{3}$$

Our key motivation is to design a unified GFlowNet to sample candidates according to different reward functions, even ones not seen during training. Instead of defining the reward function with a

fixed preference vector $\lambda$, we propose to train a preference-conditioned GFlowNet on the distribution of reward functions $R_\lambda$, where $\lambda$ is sampled from a simplex $S_M = \{\lambda \in \mathbb{R}^m | \sum_i \lambda_i = 1, \lambda_i \geq 0\}$:

$$\theta^* = \arg\min_\theta \mathbb{E}_{\lambda \in S_M} \mathbb{E}_{s \in \mathcal{S}} \mathcal{L}_\theta(s, R_\lambda) \tag{4}$$

**Remarks.** Assuming an infinite model capacity, it is easy to prove that the proposed optimization scheme (Equation (4)) is as powerful as the original one (Equation (3)), since the solutions to both loss functions coincide [17]. Nevertheless, the assumption of infinite capacity is extremely strict and hardly holds, so how to design the conditioning mechanism in practice becomes crucial.

### 4.2.1 Hypernetwork-based conditioning mechanism

We propose to condition the GFlowNets on the preference vectors via hypernetworks [26]. Hypernetworks are networks that generate the weights of a target network based on inputs. In vanilla GFlowNets, the flow predictor $F_\theta$ is parameterized with the message passing neural network (MPNN) [23] over the graph of molecular fragments, with two prediction heads approximating $F(s, s')$ and $F(s)$ based on the node and graph representations, respectively. These two heads are parameterized with multi-layer perceptrons (MLPs). For brevity, we write $\theta = (\theta_{\text{mpnn}}, \theta_{\text{pred}})$.

One can view the training of HN-GFN as learning an agent to carry out multiple policies that correspond to different goals (reward functions $R_\lambda$) defined in the same environment (state space $\mathcal{S}$ and action space $\mathcal{A}$). We thus propose only to condition the weights of prediction heads $\theta_{\text{pred}}$ with hypernetworks, while sharing the weights of MPNN $\theta_{\text{mpnn}}$, leading to more generalizable state representations. More precisely, the hypernetwork $h(\cdot; \phi)$ takes the preference vector $\lambda$ as inputs and outputs the weights $\theta_{\text{pred}} = h(\lambda; \phi)$ of prediction heads. The parameters $\phi$ of the hypernetwork are optimized like normal parameters. Following Navon et al. [47], we parametrize $h$ using an MLP with multiple heads, each generating weights for different layers of the target network.

### 4.2.2 As the acquisition function optimizer

**Training.** At each iteration, we first sample a preference vector $\lambda$ from a Dirichlet distribution $\text{Dir}(\alpha)$. Then the HN-GFN is trained with the reward function $R_\lambda(x) = a(\mu(s_\lambda(x)), \sigma(s_\lambda(x)); \mathcal{M})$, where $\mu$ and $\sigma$ are the posterior mean and standard deviation estimated by $\mathcal{M}$. In principle, we retrain HN-GFN after every round. We also tried only initializing the parameters of the hypernetwork and found it brings similar performance and is more efficient.

**Sampling.** At each round $i$, we use the trained HN-GFN to sample a diverse batch of $b$ candidates. Let $\Lambda^i$ be the set of $k$ target preference vectors $\lambda_{\text{target}}^i$. We sample $\frac{b}{k}$ molecules for each $\lambda_{\text{target}}^i \in \Lambda^i$ and evaluate them on the oracle in parallel. We simply construct $\Lambda^i$ by sampling from $\text{Dir}(\alpha)$, but it is worth noting that this prior distribution can also be defined adaptively based on the trade-off of interest. As the number of objectives increases, we choose a larger $k$ to cover the objective space.

### 4.3 Hindsight-like off-policy strategy

Resorting to the conditioning mechanism, HN-GFN can learn a family of policies to achieve various goals, i.e., one can treat sampling high-reward molecules for a particular preference vector as a separate goal. As verified empirically in Jain et al. [28], since the FM objective is *off-policy* and *offline*, we can use offline trajectories to train the target policy for better exploration, so long as the assumption of full support holds. Our key insight is that each policy can learn from the valuable experience (high-reward molecules) of other similar policies.

Inspired by hindsight experience replay [2] in RL, we propose to share high-performing molecules among policies by re-examining them with different preference vectors. As there are infinite possible preference vectors, we focus on target preference vectors $\Lambda^i$, which are based on to sample candidates, and build a replay buffer for each $\lambda_{\text{target}}^i \in \Lambda^i$. After sampling some trajectories during training, we store in the replay buffers the complete objects $x$ with the corresponding reward $R_{\lambda_{\text{target}}^i}(x)$.

To be specific, at each iteration, we first sample a preference vector from a mixture between $\text{Dir}(\alpha)$ and a uniform distribution over $\Lambda^i$: $(1 - \gamma)\text{Dir}(\alpha) + \gamma\text{Uniform}$. If $\Lambda^i$ is chosen, we construct half of the training batch with offline trajectories from the corresponding replay buffer of molecules

encountered with the highest rewards. Otherwise, we incorporate offline trajectories from the current observed dataset $\mathcal{D}_i$ instead to ensure that HN-GFN samples correctly in the vicinity of the observed Pareto set. Our strategy allows for flexible trade-offs between generalization and specialization. As we vary $\gamma$ from 0 to 1, the training distribution of preference vectors moves from $\text{Dir}(\alpha)$ to $\Lambda^i$. Exclusively training the HN-GFN with the finite target preference vectors $\Lambda^i$ can lead to poor generalization. In practice, although we only sample candidates based on $\Lambda^i$, we argue that it is vital to keep the generalization to leverage the trained HN-GFN to explore various preference vectors adaptively. The detailed algorithm can be found in Appendix A.

### 4.4 Surrogate model and acquisition function

While GPs are well-established in continuous spaces, they scale poorly with the number of observations and do not perform well in discrete spaces [57]. There has been significant work in efficiently training non-Bayesian neural networks to estimate epistemic uncertainty [21, 39]. We benchmark widely used approaches (in Appendix C.3), and use a flexible and effective one: evidential deep learning [1]. As for the acquisition function, we use Upper Confidence Bound [55] to incorporate the uncertainty. To be precise, the objectives are modeled with a single multi-task MPNN, and the acquisition function is applied to the scalarization.

## 5 Experiments

We first verify that HN-GFN has adequate capacity to generalize over preference vectors in a single-round synthetic scenario. Next, we evaluate the effectiveness of the proposed MOBO algorithm based on HN-GFN in multi-round practical scenarios, which are more in line with real-world molecular optimization. Besides, we conduct several ablation studies to empirically justify the design choices.

### 5.1 Single-round synthetic scenario

Here, our goal is to demonstrate that we can leverage the HN-GFN to sample molecules with preference-conditioned property distributions. The HN-GFN is used as a stand-alone optimizer outside of MOBO to directly optimize the scalarizations of oracle scores. As the oracle cannot be called as many times as necessary practically, we refer to this scenario as a synthetic scenario. To better visualize the trend of the property distribution of the sampled molecules as a function of the preference weight, we only consider two objectives: inhibition scores against glycogen synthase kinase-3 beta (GNK3$\beta$) and c-Jun N-terminal kinase-3 (JNK3) [40, 33].

**Compared methods.** We compare HN-GFN against the following methods. Preference-specific GFlowNets (**PS-GFN**) is a set of vanilla unconditional GFlowNets, each trained separately for a specific preference vector. Note that PS-GFN is treated as "*gold standard*" rather than a baseline, as it is trained and evaluated using the same preference vector. **Concat-GFN** and **FiLM-GFN** are two alternative conditional variations of GFlowNet based on the concatenation and FiLM [51], respectively. **MOEA/D** [64] and **NSGA-III** [14] are two multi-objective evolutionary algorithms that also incorporate preference information. We perform evolutionary algorithms over the 32-dim latent space learned by HierVAE [32], which gives better optimization performance than JT-VAE [31].

**Metrics.** All the above methods are evaluated over the same set of 5 evenly spaced preference vectors. For each GFlowNet-based method, we sample 1000 molecules per preference vector as solutions. We compare the aforementioned methods on the following metrics: **Hypervolume indicator (HV)** measures the volume of the space dominated by the Pareto front of the solutions and bounded from below by the preference point $(0, 0)$. **Diversity (Div)** is the average pairwise Tanimoto distance over Morgan fingerprints. **Correlation (Cor)** is the Spearman's rank correlation coefficient between the probability of sampling molecules from an external test set under the GFlowNet and their respective rewards in the logarithmic domain [48]. See more details in Appendix B.1.2. In a nutshell, HV and Div measure the quality of the solutions, while Cor measures how well the trained model is aligned with the given preference vector. Each experiment is repeated with 3 random seeds.

**Experimental results.** As shown in Table 1, HN-GFN outperforms the baselines and achieves competitive performance to PS-GFN (*gold standard*) on all the metrics. Compared to GFlowNet-based

Table 1: Evaluation of different methods on the synthetic scenario.

| Method | HV (↑) | Div (↑) | Cor (↑) |
|--------|--------|---------|---------|
| MOEA/D | 0.182 ± 0.045 | n/a | n/a |
| NSGA-III | 0.364 ± 0.041 | n/a | n/a |
| PS-GFN | 0.545 ± 0.055 | 0.786 ± 0.013 | 0.653 ± 0.003 |
| Concat-GFN | 0.534 ± 0.069 | 0.786 ± 0.004 | 0.646 ± 0.008 |
| FiLM-GFN | 0.431 ± 0.045 | 0.795 ± 0.014 | 0.633 ± 0.009 |
| HN-GFN | **0.550 ± 0.074** | **0.797 ± 0.015** | **0.666 ± 0.010** |

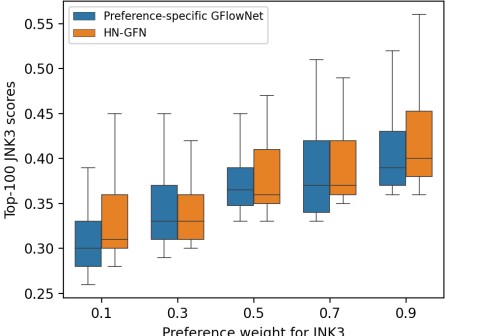 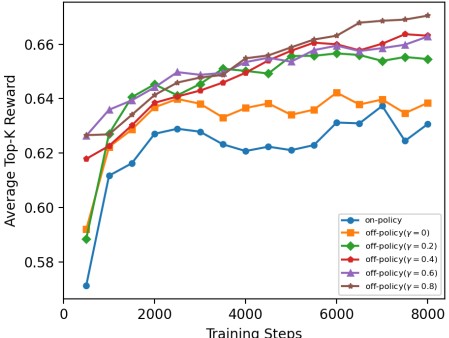

Figure 2: **Left**: The distribution of Top-100 JNK3 scores. **Right**: The progression of the average Top-20 rewards over the course of training of the HN-GFN in optimizing GSK3$\beta$ and JNK3.

methods, evolutionary algorithms (MOEA/D and NSGA-III) fail to find high-scoring molecules, especially the MOEA/D. HN-GFN outperforms Concat-GFN and FiLM-GFN, and is the only conditional variant that can match the performance of PS-GFN, implying the superiority of the well-designed hypernetwork-based conditioning mechanism. In Figure 2 (left), we visualize the empirical property (JNK3) distribution of the molecules sampled by HN-GFN and PS-GFN conditioned on the set of evaluation preference vectors. We observe that the distributions are similar and the trends are consistent: the larger the preference weight, the higher the average score. The comparable performance and consistent sampling distributions illustrate that HN-GFN has adequate capacity to generalize over preference vectors. Since the runtime and storage space of PS-GFN scale linearly with the number of preference vectors, our unified HN-GFN provides a significantly efficient way to explore various trade-offs between objectives.

## 5.2 Multi-objective Bayesian optimization

Next, we evaluate the effectiveness of HN-GFN as an acquisition function optimizer within MOBO in practical scenarios. We consider the following objective combinations of varying sizes:

- GNK3$\beta$+JNK3: Jointly inhibiting Alzheimer-related targets GNK3$\beta$ and JNK3.
- GNK3$\beta$+JNK3+QED+SA: Jointly inhibiting GNK3$\beta$ and JNK3 while being drug-like and easy-to-synthesize.

We rescale the SA score (initially between 10 and 1) so that all the above properties have a range of [0,1] and higher is better. For both combinations, we consider starting with $|\mathcal{D}_0| = 200$ random molecules and further querying the oracle $N = 8$ rounds with batch size $b = 100$. Each experiment is repeated with 3 random seeds.

**Baselines.** We compare HN-GFN with three representative LSO methods (*q***ParEGO** [37], *q***EHVI** [12], and **LaMOO** [65]), as well as a variety of state-of-the-art combinatorial optimization

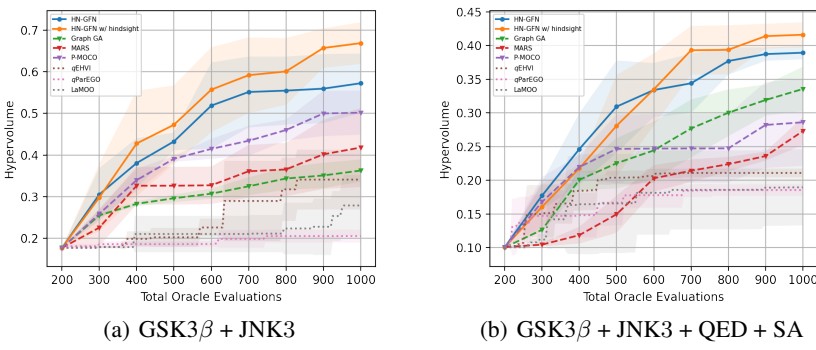

| (a) GSK3$\beta$ + JNK3 | (b) GSK3$\beta$ + JNK3 + QED + SA |

Figure 3: Optimization performance (hypervolume) over MOBO loops.

Table 2: Diversity for different methods in MOBO scenarios.

|  | **Div** ($\uparrow$) | |
|  | GSK3$\beta$ + JNK3 | GSK3$\beta$ + JNK3 + QED + SA |
|---|---|---|
| Graph GA | 0.347 ± 0.059 | 0.562 ± 0.031 |
| MARS | 0.653 ± 0.072 | **0.754 ± 0.027** |
| P-MOCO | 0.646 ± 0.008 | 0.350 ± 0.130 |
| HN-GFN | **0.810 ± 0.003** | 0.744 ± 0.008 |
| HN-GFN w/ hindsight | 0.793 ± 0.007 | 0.738 ± 0.009 |

methods: **Graph GA** [30] is a genetic algorithm, **MARS** [61] is a MCMC sampling approach, and **P-MOCO** [41] is a multi-objective RL method. We provide more details in Appendix B.2.

**Experimental results.** Figure 3 illustrates that HN-GFN achieves significantly superior performance (HV) to baselines, especially when trained with the proposed hindsight-like off-policy strategy. HN-GFN outperforms the best baselines P-MOCO and Graph GA of the two objective combinations by a large margin (0.67 vs. 0.50 and 0.42 vs. 0.34), respectively. Besides, HN-GFN is more sample-efficient, matching the performance of baselines with only half the number of oracle evaluations. All combinatorial optimization methods result in more performance gains compared to the LSO methods, implying that it is promising to optimize directly over the discrete space. In Table 2, we report the Div (computed among the batch of 100 candidates and averaged over rounds). For a fair comparison, we omit the LSO methods as they only support 160 rounds with batch size 5 due to memory constraints. Compared with Graph GA and P-MOCO, which sometimes propose quite similar candidates, the superior optimization performance of HN-GFN can be attributed to the ability to sample a diverse batch of candidates. Another interesting observation, in the more challenging setting (GNK3$\beta$+JNK3+QED+SA), is that MARS generates diverse candidates via MCMC sampling but fails to find a high-quality Pareto front, indicating that HN-GFN can find high-reward modes better than MARS. The computational costs are discussed in Appendix B.4.

### 5.3 Ablations

**Effect of the hindsight-like off-policy strategy.** In the first round of MOBO, for each $\lambda_{\text{target}} \in \Lambda$ we sample 100 molecules every 500 training steps and compute the average Top-20 reward over $\Lambda$. In Figure 2 (right), we find that the hindsight-like off-policy strategy significantly boosts average rewards, demonstrating that sharing high-performing molecules among policies effectively speeds up the training of HN-GFN. On the other hand, further increasing $\gamma$ leads to slight improvement. Hence, we choose $\gamma = 0.2$ for the desired trade-off between generalization and specialization.

**Effect of** $\mathrm{Dir}(\alpha)$**.** Here we consider the more challenging GNK3$\beta$+JNK3+QED+SA combination, where the difficulty of optimization varies widely for various properties. Table 3 shows that the distribution skewed toward harder properties results in better optimization performance. In our early experiments, we found that if the distribution of $\Lambda^i$ is fixed, HN-GFN is quite robust to changes in $\alpha$.

**Effect of scalarization functions** $s_\lambda$**.** In Table 3, we observe that WS leads to a better Pareto front than Tchebycheff. Although the simple WS is generally inappropriate when a non-convex Pareto front is encountered [10], we find that it is effective when optimized with HN-GFN, which can sample diverse high-reward candidates and may reach the non-convex part of the Pareto front. In addition, we conjecture that the non-smooth reward landscapes induced by Tchebycheff are harder to optimize.

Table 3: Ablation study of the $\alpha$ and scalarization functions on GNK3$\beta$+JNK3+QED+SA.

| | $\alpha$ | | | scalarization function | |
|---|---|---|---|---|---|
| | (1,1,1,1) | (3,3,1,1) | (3,4,2,1) | WS | Tchebycheff |
| HV | 0.312 ± 0.039 | 0.385 ± 0.018 | **0.416 ± 0.023** | **0.416 ± 0.023** | 0.304 ± 0.075 |
| Div | **0.815 ± 0.015** | 0.758 ± 0.018 | 0.738 ± 0.009 | **0.738 ± 0.009** | 0.732 ± 0.014 |

# 6 Conclusion

We have introduced a MOBO algorithm for sample-efficient multi-objective molecular optimization. This algorithm leverages a hypernetwork-based GFlowNet (HN-GFN) to sample a diverse batch of candidates from the approximate Pareto front. In addition, we present a hindsight-like off-policy strategy to improve optimization performance. Our algorithm outperforms existing approaches in synthetic and practical scenarios. **Future work** includes extending this algorithm to other discrete optimization problems such as biological sequence design and neural architecture search. **One limitation** of the proposed HN-GFN is the higher computational costs than other training-free optimization methods. However, the costs resulting from model training are negligible compared to the costs of evaluating the candidates in real-world applications. We argue that the higher quality of the candidates is much more essential than lower costs.

## Acknowledgments and Disclosure of Funding

This research was partially supported by National Key R&D Program of China under grant No.2018AAA0102102, National Natural Science Foundation of China under grants No.62176231, No.62106218, No.82202984, No.92259202, No.62132017 and U22B2034, Zhejiang Key R&D Program of China under grant No.2023C03053. Tingjun Hou's research was supported in part by National Natural Science Foundation of China under grant No.22220102001.

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

# A Algorithms

Algorithm 1 describes the overall framework of the proposed MOBO algorithm, where HN-GFN is leveraged as the acquisition function optimizer. Algorithm 2 describes the training procedure for HN-GFN within MOBO.

---

**Algorithm 1** MOBO based on HN-GFN

---

**Input:** oracle $f = (f_1, \ldots, f_M)$, initial dataset $\mathcal{D}_0 = \{(x_i^0, f(x_i^0))\}_{i=1}^n$, acquisition function $a$, parameter of Dirichlet distribution $\alpha$, number of rounds $N$, batch size $b$
**Initialization:** surrogate model $\mathcal{M}$, parameters of HN-GFN $\pi_\theta$
**for** $i = 1$ **to** $N$ **do**
    Fit surrogate model $\mathcal{M}$ on dataset $\mathcal{D}_{i-1}$
    Sample the set of target preference weights $\Lambda \sim \text{Dir}(\alpha)$
    Train $\pi_\theta$ with reward function $R_\lambda(x) = a(\mu(s_\lambda(x)), \sigma(s_\lambda(x)); \mathcal{M})$      ▷ Algorithm 2
    Sample query batch $\mathcal{B}_i = \{x_j^i\}_{j=1}^b$ based on $\lambda_{target} \in \Lambda$
    Evaluate batch $\mathcal{B}_i$ with $f$ and augment the dataset $\mathcal{D}_{i+1} = \mathcal{D}_i \cup \{(x_j^i, f(x_j^i))\}_{j=1}^b$
**end for**

---

---

**Algorithm 2** Training procedure for HN-GFN with the hindsight-like off-policy strategy

---

**Input:** available dataset $\mathcal{D}_i$, reward function $R$, minibatch size $m$, set of target preference vectors $\Lambda$, proportion of hindsight-like strategy $\gamma$, replay buffers $\{\mathcal{R}_\lambda\}_{\lambda \in \Lambda}$
**while** not converged **do**
    Flag $\sim$ Bernoulli$(\gamma)$
    **if** Flag $= 1$ **then**
        $\lambda \sim \Lambda$
        Sample $\frac{m}{2}$ trajectories from replay buffer $\mathcal{R}_\lambda$
    **else**
        $\lambda \sim \text{Dir}(\alpha)$
        Sample $\frac{m}{2}$ trajectories from the available dataset $\mathcal{D}_i$
    **end if**
    $\theta = (\theta_{\text{mpnn}}, h(\lambda; \phi))$
    Sample $\frac{m}{2}$ trajectories from policy $\widetilde{\pi}$ and store terminal states $x$ in $\mathcal{R}_\lambda$ for all $\lambda \in \Lambda$
    Compute reward $R_\lambda(x)$ on terminal states $x$ from each trajectory in the minibatch
    Update parameters $\theta_{\text{mpnn}}$ and $\phi$ with a stochastic gradient descent step w.r.t Equation (4)
**end while**

---

# B Implementation details

## B.1 Experimental settings

### B.1.1 Molecule domain

Following Bengio et al. [6], the molecules are generated by sequentially attaching a fragment, which is chosen from a predefined vocabulary of building blocks, to an atom of the partially constructed molecules. The maximum trajectory length is 8, with the number of actions varying between 100 and 2000 depending on the state, making $|\mathcal{X}|$ up to $10^{16}$. We adopt the property prediction models released by Xie et al. [61] to evaluate the inhibition ability of generated molecules against GSK3$\beta$ and JNK3.

### B.1.2 Metrics

**Diversity.** Diversity (Div) is the average pairwise Tanimoto distance over Morgan fingerprints. In the synthetic scenario, for each preference vector, we sample 1000 molecules, calculate the Div among the Top-100 molecules, and report the averages over preferences. In MOBO, the DiV is computed among the batch of 100 candidates per round, as Graph GA and MARS are not preference-conditioned. And we believe this metric possibly is more aligned with how these methods might be used in a biology or chemistry experiment.

**Correlation.** Correlation (Cor) is the Spearman's rank correlation coefficient between the probability of sampling molecules from an external test set under the GFlowNet and their respective rewards in the logarithmic domain: Cor = Spearman's $\rho_{\log(\pi(x)),\log(R(x))}$. The external test set is obtained in two steps: First, we generate a random dataset containing 300K molecules uniformly based on the number of building blocks; Next, we sample the test sets with uniform property distribution corresponding to GSK3$\beta$ and JNK3, respectively, from the 300K molecules.

## B.2 Baselines

All baselines are implemented using the publicly released source codes with adaptations for our MOBO scenarios. The evolutionary algorithms (MOEA/D and NSGA-III) are implemented in PyMOO [9], and the LSO methods (qParEGO, qEHVI, and LaMOO) are implemented in BoTorch [4]. In MOBO scenarios, LaMOO and Graph GA utilize EHVI as the acquisition function. For all GP-based methods, each objective is modeled by an independent GP.

## B.3 HN-GFN

We implement the proposed HN-GFN in PyTorch [50]. The values of key hyper-parameters are illustrated in Table 4.

**Surrogate model:** We use the 12-layer MPNN as the base architecture of the surrogate model in our experiments. In MOBO, a single multi-task MPNN is trained with a batch size of 64 using the Adam optimizer with a dropout rate of 0.1 and a weight decay rate of 1e-6. We apply early stopping to improve generalization.

**HN-GFN:** HN-GFN contains a vanilla GFlowNet and a preference-conditioned hypernetwork. The architecture of GFlowNet is a 10-layer MPNN, and the hypernetwork is a 3-layer MLP with multiple heads, each generating weights for different layers of the target network. The HN-GFN is trained with Adam optimizer to optimize the Flow Matching objective.

Table 4: Hyper-parameters used in the real-world MOBO experiments.

| Hyper-parameter | GSK3$\beta$ + JNK3 | GSK3$\beta$ + JNK3 + QED + SA |
|---|---|---|
| *Surrogate model* | | |
| Hidden size | 64 | 64 |
| Learning rate | 2.5e-4 | 1e-3 |
| $\lambda$ for evidential regression | 0.1 | 0.1 |
| Number of iterations | 10000 | 10000 |
| Early stop patience | 500 | 500 |
| Dropout | 0.1 | 0.1 |
| Weight decay | 1e-6 | 1e-6 |
| *Acquisition function (UCB)* | | |
| $\beta$ | 0.1 | 0.1 |
| *HN-GFN* | | |
| Learning rate | 5e-4 | 5e-4 |
| Reward exponent | 8 | 8 |
| Reward norm | 1.0 | 1.0 |
| Trajectories minibatch size | 8 | 8 |
| Offline minibatch size | 8 | 8 |
| hindsight $\gamma$ | 0.2 | 0.2 |
| Uniform policy coefficient | 0.05 | 0.05 |
| Hidden size for GFlowNet | 256 | 256 |
| Hidden size for hypernetwork | 100 | 100 |
| Training steps | 5000 | 5000 |
| $\alpha$ | (1,1) | (3,4,2,1) |

### B.4 Empirical running time

The efficiency is compared on the same computing facilities using 1 Tesla V100 GPU. In the context of MOBO, the running time of three LSO methods (i.e., qParEGO, qEHVI, and LaMOO) is around 3 hours, while Graph GA optimizes much faster and costs only 13 minutes. In contrast, the time complexity of deep-learning-based discrete optimization methods is much larger. MARS costs 32 hours, while our proposed HN-GFN costs 10 hours. With the hindsight-like training strategy, the running time of HN-GFN will increase roughly by 33%.

However, if we look at the problem in a bigger picture, the time costs for model training are likely negligible compared to those of evaluating molecular candidates in real-world applications. Therefore, we argue that the high quality of the candidates (the performance of the MOBO algorithm) is more essential than having a lower training cost.

## C Additional results

### C.1 Sampled molecules in MOBO experiments

We give some examples of sampled molecules from the Pareto front by HN-GFN in the GSK3$\beta$ + JNK3 + QED + SA optimization setting (Figure 4). The numbers below each molecule refer to GSK3$\beta$, JNK3, QED, and SA scores respectively.

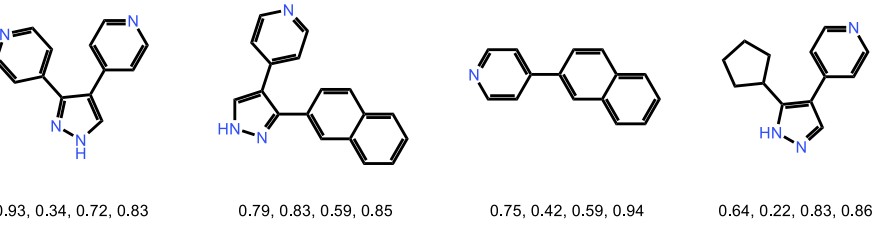

| 0.93, 0.34, 0.72, 0.83 | 0.79, 0.83, 0.59, 0.85 | 0.75, 0.42, 0.59, 0.94 | 0.64, 0.22, 0.83, 0.86 |

Figure 4: Sampled molecules from the approximate Pareto front by HN-GFN.

### C.2 Synthetic scenario

As illustrated in Figure 5, the distribution of Top-100 GSK3$\beta$ scores shows a consistent trend in preference-specific GFlowNet and our proposed HN-GFN, although the trend is not as significant as the JNK3 property.

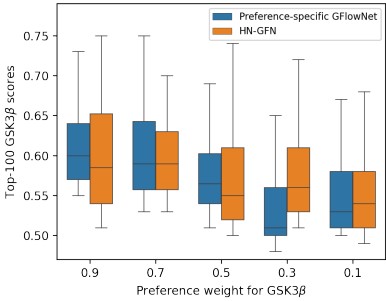

Figure 5: Comparison of the distribution of Top-100 GSK3$\beta$ scores sampled by different preference vectors using preference-specific GFlowNets and HN-GFN.

## C.3  Effect of Surrogate models

We conduct ablation experiments to study the effectiveness of different surrogate models. We consider the following three surrogate models: evidential regression [1], Deep Ensembles [39], and GP based on the Tanimoto kernel [59]. As shown in Table 5, we can observe that evidential regression leads to better optimization performance than Deep Ensembles. While the HV of evidential regression and GP is comparable, evidential regression can propose more diverse candidates. Furthermore, we argue that GP is less flexible over discrete spaces than evidential regression and Deep Ensembles, as different kernels need to be designed according to the data structures.

Table 5: Evaluation of different surrogate models in MOBO scenarios

|  | GSK3$\beta$ + JNK3 | | GSK3$\beta$ + JNK3 + QED + SA | |
| --- | --- | --- | --- | --- |
|  | HV | Div | HV | Div |
| HN-GFN (Evidential) | **0.669 ± 0.061** | 0.793 ± 0.007 | 0.416 ± 0.023 | 0.738 ± 0.009 |
| HN-GFN (Ensemble) | 0.583 ± 0.103 | **0.797 ± 0.004** | 0.355 ± 0.048 | **0.761 ± 0.012** |
| HN-GFN (GP) | 0.662 ± 0.054 | 0.739 ± 0.008 | **0.421 ± 0.037** | 0.683 ± 0.018 |

## D  Comparison with parallel work

While the concept of Pareto GFlowNet was theoretically discussed in Bengio et al. [7], we are among the first to study and instantiate this concept for MOO, and we address practical challenges that are not discussed thoroughly in the original theoretical exposition to potentially make sample-efficient molecular optimization a reality. We extensively study the impact of the conditioning mechanism (Section 5.1) and surrogate models (Appendix C.3). Moreover, we delicately propose a hindsight-like off-policy strategy (Section 4.3) which is rarely studied for MOO (in both the RL and GFlowNet literature).

We note that there is a parallel work introduced in Jain et al. [29], which shares the idea of using GFlowNets for MOO. In Jain et al. [29], they propose two variants MOGFN-PC and MOGFN-AL for MOO in the single-round scenario and Bayesian optimization (BO) scenario, respectively. Indeed, when HN-GFN is used as a stand-alone optimizer outside of MOBO (Section 5.1), it is similar to MOGFN-PC except for using different conditioning mechanisms. As for MOBO, MOGFN-AL is a vanilla GFlowNet whose reward function is defined as a multi-objective acquisition function (NEHVI [13]). In addition to extending GFlowNet, we delicately propose a hindsight-like off-policy strategy (Section 4.3) which is rarely studied for MOO (in both the RL and GFlowNet literature).

