# OpenReview forum: "Sample-efficient Multi-objective Molecular Optimization with GFlowNets"
_NeurIPS.cc/2023/Conference — NeurIPS 2023 poster_

### Official Review · Reviewer_Dc63 · 2023-06-15

**Soundness:** 3 good
**Presentation:** 2 fair
**Contribution:** 3 good
**Rating:** 6
**Confidence:** 3

**Summary:**

Molecule generation involves the optimization of multiple potentially competing objectives simultaneously. As evaluating these objectives can be a time-consuming and costly task, sample efficiency is paramount. This work proposes a multi-objective Bayesian optimization approach leveraging GFlowNets to tackle this problem. The authors propose a preference-based decomposition and a hyper-network-based parameterization for incorporating preferences and conditioning on those. They also employ a hindsight-experience replay strategy to utilize offline data during learning. The hyper-network-based GFlowNet is then used as an acquisition function optimizer to find a diverse set of molecules on the empirical Pareto front.

**Strengths:**

- The paper covers an important and impactful topic of multi-objective optimization in molecule design.
-  The experimental results, in principle, are strong.
- It is an original combination of existing techniques.
- Many benchmarks are compared.
- Ablation study

**Weaknesses:**

- It is not directly evident to me how this work is different from Pareto GFlowNet that was proposed in [GFlowNet Foundations](https://arxiv.org/pdf/2111.09266.pdf)

- Some parts remain somewhat unclear to me. How is the hyper-network trained? Is it stable, does it do what it is expected to do?

- I am somewhat missing the link between the surrogate model and HN-GFN from what is described in the main paper. Why do we actually need the surrogate model? Can't the GFlownet propose diverse candidates by itself?

- Perhaps the work would benefit from a clearer description and/or toy tasks to clear this up.

**Questions:**

- It would be nice if the authors could dedicate some more information about the test problem. What is the search space of the molecules, what atom types, how are impossible molecular structures avoided, etc? Are the building blocks created in such a way that they always can be connected to each other without violations?

- Why a batch size of 100 and 8 rounds? In what setting would a practitioner be able to perform 100 wet lab experiments simultaneously? Perhaps with a simulator, this would be possible. But wouldn't the surrogate models (at least from the BO qEHVI and qParEGO) benefit from lower batch sizes and more iterations? This could change the benchmarks significantly.

**Limitations:**

The authors have addressed limitations.

---

> ### Author Rebuttal · Authors · 2023-08-10
>
> We deeply appreciate the reviewer for the insightful and constructive comments!
>
> > It is not directly evident to me how this work is different from Pareto GFN proposed in GFlowNet Foundations.
>
> While the concept of Pareto GFN was theoretically discussed in GFlowNet Foundations, we are among the first to study and instantiate this concept for MOO, and we address practical challenges (not discussed thoroughly in the original theoretical exposition) to potentially make sample-efficient molecular optimization a reality. We extensively study the impact of the conditioning mechanism (Sec5.1) and surrogate models (Sec4.4). Moreover, we delicately propose a hindsight-like off-policy strategy (Sec4.3) which is rarely studied for MOO (in both RL and GFN literature).
>
> > How is the hypernetwork trained? Is it stable, does it do what it is expected to do?
>
> We are sorry for our unclear description. Hypernetworks generate the weights of a target network based on inputs. In our implementations, the hypernetwork takes the preference vector $λ$ as inputs and outputs the weights of prediction heads upon the MPNN encoder. In short, the parameters of the hypernetwork are optimized like normal parameters, while the parameters of the prediction heads are the output of the hypernetwork.
>
> In our experiments, the hypernetwork is stable and does what it is expected to do. We attribute this to our design where only the weights of prediction heads are conditioned with hypernetwork while the weights of the encoder are shared among all preference vectors (Sec4.2.1). We study various conditioning mechanisms and find that the hypernetwork-based approach performs best.
>
> > I am somewhat missing the link between the surrogate model and HN-GFN. Why do we actually need the surrogate model? Can't GFN propose diverse candidates by itself? Perhaps the work would benefit from a clearer description and/or toy tasks to clear this up.
>
> We apologize for our unclear description. We propose to extend GFN for MOO. We first consider a synthetic single-round scenario (Sec5.1) where the oracle is assumed to be called as many times as necessary, without the need of a surrogate model. In this case, GFN can directly optimize the reward defined by the oracle. Practically, realistic oracles are extremely expensive to evaluate. We thus focus more on the BO scenario (Sec5.2) where the oracle is given an evaluation budget. Then we need a cheap statistic surrogate model to approximate the expensive oracle. As the surrogate model cannot exactly reproduce the oracle's full behaviors, an ideal one should provide a calibrated uncertainty (exploration) estimate. By incorporating uncertainty into the reward function, GFNs search not only the space where the model prediction is high (exploitation) but also the space where the uncertainty is large (exploration). This behavior is paramount for BO.
>
> > It would be nice if the authors could dedicate some more information about the test problem. What is the search space of the molecules, what atom types, how are impossible molecular structures avoided, etc? Are the building blocks created in such a way that they always can be connected to each other without violations?
>
> We appreciate the reviewer's helpful reminder. We follow the molecule environment of [1], where the molecules are generated by sequentially attaching a fragment (from a predefined vocabulary of building blocks) to an atom of the partially constructed molecules. The maximum trajectory length is 8, with the number of actions varying between 100 and 2000 depending on the state (the larger a molecule, the more possible additions exist), making $|\mathcal{X}|$ up to $10^{16}$. The blocks are chosen via the process proposed in JT-VAE [2]. To ensure valid molecular structures, we only predict action (which block to attach) for each stem of the graph (stems are atoms which new blocks can be attached to). We will revise our writing to dedicate experimental settings.
>
> > Why a batch size of 100 and 8 rounds? In what setting would a practitioner be able to perform 100 wet lab experiments simultaneously? Perhaps with a simulator, this would be possible. But wouldn't the surrogate models (at least from qEHVI and qParEGO) benefit from lower batch sizes and more iterations? This could change the benchmarks significantly.
>
> While these choices are unusual for common problems in BO, they are common but challenging in molecular optimization and biological sequence design, where large-batch and low-round setting is desired, as candidates can be synthesized and evaluated in parallel in biochemical experiments. For example, DyNA PPO [3] considers 10 rounds with a batch size of 100. The original paper of GFlowNet [1] even considers a batch size of 200.
>
> I agree that the surrogate models benefit from lower batch sizes and more iterations. However, in practice, the time costs for proposing candidates are negligible compared to evaluation, especially when the candidates are proposed by generative models, these novel candidate molecules may not be available for purchase. Customized synthesis may take weeks if not more for each round; hence we consider such an experimental setting to be more practical in chemistry-related fields. As we noted in line 322, the latent space optimization (LSO) methods (qEHVI, qParEGO, and LaMOO) only support 160 rounds with batch size 5 due to memory constraints. Nevertheless, LSO methods still fall far short of combinatorial optimization methods. We believe that the reason is that the latent space learned by generative models (HierVAE and JT-VAE) is not sufficiently expressive and discriminative.
>
> [1] Flow network based generative models for non-iterative diverse candidate generation
>
> [2] Junction tree variational autoencoder for molecular graph generation
>
> [3] Model-based reinforcement learning for biological sequence design
>
> **We hope our response can alleviate your concerns. Please let us know if you have any additional questions.**

---

> > ### Comment · Reviewer_Dc63 · 2023-08-17
> >
> > Thank you for the author's comprehensive response and clarifications. That clears quite some things up for me.
> > I will increase the score to a 6.

---

### Official Review · Reviewer_hutP · 2023-07-04

**Soundness:** 4 excellent
**Presentation:** 3 good
**Contribution:** 3 good
**Rating:** 5
**Confidence:** 5

**Summary:**

A GFlowNet for molecular optimization conditioned on the preference weights of multiple objectives is proposed. To be precise, the model is trained to sample molecules from a target space with probability proportional to some combination of reward functions (weighted sum or Chebyshev scalarization) and trained with varying weights of the rewards, with the GFlowNet conditioning achieved by a hypernetwork parametrization. The system is tested in both single-step and Bayesian optimization / active learning settings.

**Strengths:**

- The writing is generally clear, problem is introduced well and placed in context.
- Good results on the tasks studied relative to the chosen baselines
- Two interesting GFlowNet contributions not specific to molecules:
  - Study of conditioning algorithms (FiLM and Concat variants), with the hypernetwork-based approach found to perform best.
  - The use of a conditional replay buffer, which is, as far as I know, new in GFlowNet literature (a replay buffer was used with GFlowNets in [arXiv:2202.13903, UAI 2022] but in a vanilla RL manner).
  - Both are illustrated well with ablation studies.
- The code is a helpful addition that aids in understanding the details of what was done.

**Weaknesses:**

Meta-weakness: Due to my closely following the literature, I was aware of an earlier submission of this paper to ICLR 2023. The present submission is almost identical to ICLR version (post-rebuttal) and therefore cannot incorporate the feedback of the reviewers. That said, to avoid biasing myself here, I did not read the ICLR reviews and this assessment is entirely my own.

Comparison with prior work:
- There is the paper "Multi-objective GFlowNets" [MO; arXiv:2210.12765, ICML 2023], which also studies GFlowNets for multiobjective Bayesian optimization by conditioning on scalarization weights. Can you comment on the differences in problem setting or method with that paper?
- The attached code includes the implementations of two baselines, in `code/generator/gfn.py`, but they are not included in the paper:
  - MOReinforce, which is a natural candidate for an RL-based baseline that can readily share the parametrization of the agent with the GFlowNet.
  - GFlowNet with trajectory balance [TB; arXiv:2201.13259, NeurIPS 2022]. Trajectory balance is the objective used in nearly all work on GFlowNets for molecular optimization after [5], but for some reason it is not evaluated or mentioned in the text. In particular, it is evaluated on molecule synthesis in both [TB] and [MO].

Small issue in presentation/math: There are some inconsistencies in definitions related to GFlowNets. Equation (1) writes $R(s)$ for the reward received when terminating at $s$, as in [5]. However, the text immediately following it talks about terminal and nonterminal states (with nonterminals having reward 0), and the text above says there is a special termination action that leads to terminal states. This is more consistent with the convention from [TB], where terminal states have no children and are the only states with nonzero reward. This is not a major bug as the two conventions are equivalent, but it would be good to be careful.

**Questions:**

Please see above.

**Limitations:**

yes

---

> ### Author Rebuttal · Authors · 2023-08-09
>
> We deeply appreciate the reviewer for the insightful and constructive comments!
>
> > There is the paper "Multi-objective GFlowNets" [MO; arXiv:2210.12765, ICML 2023], which also studies GFlowNets for multiobjective Bayesian optimization by conditioning on scalarization weights. Can you comment on the differences in problem setting or method with that paper?
>
> Our work is actually concurrent with [MO]. They propose two variants MOGFN-PC and MOGFN-AL for MOO in the single-round scenario and Bayesian optimization (BO) scenario, respectively. Indeed, when HN-GFN is used as a stand-alone optimizer outside of MOBO (Section 5.1), it is similar to MOGFN-PC except for using different conditioning mechanisms. As for MOBO, MOGFN-AL is a vanilla GFlowNet whose reward function is defined as a multi-objective acquisition function (NEHVI). In our early experiments, we also tried to directly optimize this acquisition function. Unfortunately, we found that this approach is ineffective as the number of objectives increases. Because the value of NEHVI is so tiny that the policy cannot be effectively optimized. This phenomenon is tougher when the exponent of the reward is large. While in [MO], they only consider two objectives, and we consider setting up to four objectives which exposes the aforementioned challenges that we address in this work. In addition to extending GFlowNet, we delicately propose a hindsight-like off-policy strategy which is rarely studied for MOO (in both RL and GFlowNet literature).
>
> > The attached code includes the implementations of two baselines, in code/generator/gfn.py, but they are not included in the paper:
> > - MOReinforce, which is a natural candidate for an RL-based baseline that can readily share the parametrization of the agent with the GFlowNet.
> > - GFlowNet with trajectory balance [TB; arXiv:2201.13259, NeurIPS 2022]. Trajectory balance is the objective used in nearly all work on GFlowNets for molecular optimization after [5], but for some reason, it is not evaluated or mentioned in the text. In particular, it is evaluated on molecule synthesis in both [TB] and [MO].
>
> Thanks for the careful review of our code. 1) Indeed, MOReinforce (arXiv: 2203.15386, ICLR 2022) was included in the paper (referred to as P-MOCO). We believe this misunderstanding is caused by inconsistent naming: [P-MOCO] is termed MOReinforce in [MO], while we use the original name P-MOCO. 2) In our early experiments, we found that TB performs worse than FM, and we encountered a numerical issue with NaN in the predicted logits. We contacted the author of [TB] and described the situation, and he later confirmed that 'I did also get the error you reported when I ran the code myself.' Hence, we have chosen to use FM. In [MO], they did evaluate it on molecule synthesis. However, in their ablation experiments about the fragment-based molecular generation task (Figure 3c in ICML version and Table 6 in 2210.12765v1), we found that only when an extremely large exponent (96) is set can MOGFN-PC outperforms baselines (Table 2 in ICML version and Table 3 in 2210.12765v1). To some extent, we argue that setting such a large exponent is unreasonable, especially for sampling algorithms like GFlowNets, because the probability of sampling the molecules with slightly lower rewards (before the exponential operation) will be very low. Put differently, the sampling distribution is concentrated close to the modes, and the diversity of the candidates is sacrificed.
>
> > There are some inconsistencies in definitions related to GFlowNets. Equation (1) writes $R(s)$ for the reward received when terminating at $s$ as in [5]. However, the text immediately following it talks about terminal and nonterminal states (with nonterminals having reward 0), and the text above says there is a special termination action that leads to terminal states. This is more consistent with the convention from [TB], where terminal states have no children and are the only states with nonzero rewards. This is not a major bug as the two conventions are equivalent, but it would be good to be careful.
>
> Thanks again for your patience and helpful reminder. We will carefully revise these inconsistencies in definitions to ease the understanding of GFlowNets.
>
> **We hope our response can alleviate your concerns. Please let us know if you have any additional questions.**

---

> > ### Comment · Reviewer_hutP · 2023-08-10
> > **Response to rebuttal**
> >
> > Understood about baselines. I hope that this discussion will be added to the revised version of the paper.
> >
> > I am also aware of the NaN-in-logits issue with TB. I believe that past work reported the performance at the last state before NaN, but expect this would happen earlier with a higher reward exponent.
> >
> > Overall, I am satisfied with the response; having read the other reviews and responses, I maintain my score. Thank you again for the clarifications!

---

### Official Review · Reviewer_BkDd · 2023-07-06

**Soundness:** 3 good
**Presentation:** 4 excellent
**Contribution:** 3 good
**Rating:** 6
**Confidence:** 3

**Summary:**

This paper proposes the use of GFlotNets to address the problem of sample-efficient multi-objective molecular optimization, an important problem in various scientific discovery application - such as materials design and drug discovery.
The key idea proposed in this work is to leverage hypernetwork-based GFlowNets - referred to as HN-GFN - to optimize the acquisition function for multi-objective Bayesian optimization (MOBO).
The goal is to enable efficient sampling of a diverse high-quality batch of molecular candidates from an approximate Pareto front.


**Strengths:**

The proposed hyper network-based GFlowNets as a MOBO acquisition function optimizer provides a novel and intuitive way of using GFlowNets for sampling novel molecular candidates, where a "unified" GFlowNet is trained that considers the distribution of different reward functions that correspond to different preference vectors instead of relying on a single GFlowNet for a fixed preference vector.
This may allow the resulting model, HN-GFN, to naturally explore the various trade-offs between multiple objectives that may compete with one another by adapting to varying the input preference vector.

The proposed method builds on the recently proposed and widely popular GFlowNets and extends its flexibility for multi-objective molecular optimization by incorporating a hypernetwork-based approach.
Additionally, the adoption of a hindsight-like off-policy strategy is proposed to improve the learning efficiency, and ultimately, the multi-objective molecular optimization performance.

Overall, the paper is well-organized and written in a clear manner, the proposed method is well-motivated and novel, and the performance evaluation results demonstrate the potential advantages of the proposed HN-GFN.





**Weaknesses:**

Although the batch size may significantly affect the overall computational cost as well as the optimization performance, there is no discussion on the impact of selecting a specific batch size nor any empirical evaluation based on different batch sizes.

While the evaluation results provide some preliminary evidence of the potential advantages of HN-GFN, the evaluations in the current study are limited to (virtually) a single problem: i.e., inhibition of GSK3β + JNK3 (with potential additional considerations for synthesizability and drug-likeness).
Additional examples are needed to more convincingly demonstrate the general applicability (and merits) of the proposed HN-GFN.
I suggest providing further evaluation results based on other benchmark problems often used for evaluating other generative models (e.g., JT-VAE, HierVAE, etc.)

In Figure 2, only trends for JNK3 are shown, but it would be helpful to show the optimization trends for GNK3β as well for completeness.




**Questions:**

Please see the questions and suggestions in the weaknesses section.


**Limitations:**

The conclusion section includes a very brief discussion of a limitation of the current method and suggests directions for future work.
The broader implication of the work is not explicitly discussed in the manuscript.

---

> ### Author Rebuttal · Authors · 2023-08-10
>
> We deeply appreciate the reviewer for the insightful and constructive comments!
>
> > Although the batch size may significantly affect the overall computational cost as well as the optimization performance, there is no discussion on the impact of selecting a specific batch size nor any empirical evaluation based on different batch sizes.
>
> Thanks for pointing this out. We have conducted experiments to discuss the impact of different batch sizes. Just as expected, our HN-GFN benefits from lower batch size and more rounds. As GFlowNet amortizes the cost of optimization during training, the overall computational cost is proportional to the number of rounds. From the perspective of the trade-off between computational cost and optimization performance, we believe that 8*100 is suitable for the design and benchmarking of algorithms for molecular optimization.
>
> | # Round * BS | 16 * 50 | 8 * 100 | 4 * 200 |
> | :-----| ----: | ----: | ----: |
> | HV       | 0.442 ± 0.012 (0.026 $\uparrow$)| 0.416 ± 0.023 | 0.373 ± 0.038 (0.043 $\downarrow$)|
> | Run-time | 2 $\times$    | 1 $\times$    | 0.5 $\times$  |
>
> > While the evaluation results provide some preliminary evidence of the potential advantages of HN-GFN, the evaluations in the current study are limited to (virtually) a single problem: i.e., inhibition of GSK3β + JNK3 (with potential additional considerations for synthesizability and drug-likeness). Additional examples are needed to more convincingly demonstrate the general applicability (and merits) of the proposed HN-GFN. I suggest providing further evaluation results based on other benchmark problems often used for evaluating other generative models (e.g., JT-VAE, HierVAE, etc.)
>
> Thanks for pointing this out. GSK3β + JNK3 (with potential additional considerations for synthesizability and drug-likeness) is the most widely-used benchmark for multi-objective molecular optimization since proposed in [1]. For further diverse evaluation, after reviewing the related literature, we consider the following objective combinations: dopamine type 2 receptor (DRD2) + QED + SA [2] and soluble epoxide hydrolase (sEH) + QED + SA [3]. Our HN-GFN still achieves significantly superior performance than P-MOCO (the best baseline for GSK3β + JNK3).
>
> |  | DRD2 + QED + SA | | sEH + QED + SA | |
> | :-----| ----: | ----: | ----: | ----: |
> |  | HV | Div | HV | Div |
> | P-MOCO   | 0.710 ± 0.015 | 0.209 ± 0.056 | 0.649 ± 0.020 | 0.445 ± 0.005 |
> | HN-GFN   | **0.853 ± 0.001** | **0.735 ± 0.045** | **0.694 ± 0.010** | **0.828 ± 0.002** |
>
> > In Figure 2, only trends for JNK3 are shown, but it would be helpful to show the optimization trends for GNK3β as well for completeness.
>
> We appreciate the reviewer's helpful reminder. We only show trends for JNK3 due to the space limitation. We will include the trends for GNK3β as well.
>
> [1] Multi-objective molecule generation using interpretable substructures.
>
> [2] Multi-constraint molecular generation based on conditional transformer, knowledge distillation and reinforcement learning.
>
> [3] Flow network based generative models for non-iterative diverse candidate generation.
>
> **We hope our response can alleviate your concerns. Please let us know if you have any additional questions.**

---

> > ### Comment · Reviewer_BkDd · 2023-08-18
> >
> > Thank you very much for the clarification and the additional results.
> > While the rebuttal doesn't significantly change my overall evaluation (which was already on the positive side), I have updated the score on the presentation.
> > Thanks again.

---

### Official Review · Reviewer_9YCN · 2023-07-08

**Soundness:** 3 good
**Presentation:** 3 good
**Contribution:** 2 fair
**Rating:** 5
**Confidence:** 3

**Summary:**

This work addresses molecular design with GFLOWNET, an algorithm learning a sampling policy $\pi$ proportional to the reward function, i.e. where $\pi(x) \propto R(x)$. In particular, this work tackles the essential yet under-adressed multiobjective optimization setting. They do so using a hypernetwork (conditioned on a preference vector) providing weights for the sampler.

**Strengths:**

In my opinion the presentation is clear and the motivation behind this work are nicely presented. In addition, the paper is well written and easy to follow. As a side note, its presentation of the GFN is interesting and worth the reminder to ease the understanding of the paper.
The approach is both original and sound, boosting its potential use in practical application.
Finally, the presented results are conclusive and seem to validate the intuition behind the method.

**Weaknesses:**

The main weaknesses in the paper is in my opinion in the lack of diverse experiments or explanation about them. See questions.

**Questions:**

1- A part of the protocole is unclear to me: are the generated molecules evaluated on the objective using an actual oracle or using the learned surrogate function ?

2 - How are the objectives computed ? Is it “physics” based or does it rely on learning a surrogate model ? Then, how are the generated molecules evaluated ?

3 - What is the length of the generated molecules ? Is it a hyperparameter of the model ?

4 - How sensitive are the results to the original dataset  $D_0$?  In my personal experience, Gflownets tend to be rather unstable or difficult to train due to the sparse feedback (and depends on the sequence size) and 200 initial molecules seem low to be able to learn a robust policy.

5 - Does the proposed Hindsight-like off policy strategy amounts to the proposition of [1], i.e. to hybrid the learning between generated molecules vs already generated molecules ?

6 - What are the variable predicted by the MPNN ?

7 -  Can the authors discuss their implementation choice regarding the gflow network. For instance [1] use a MLP as sampler.


[1]: Jain et al, Biological sequence designs with Gflownets.

**Limitations:**

A limitation of the paper is the lack of diversity in the experimental settings. For instance, the paper Gflownet for biological sequence design presents optimization results for several benchmark datasets such as GFP. Can the authors comment on the ability of their method to adapt to such an ‘off-line’ setting (which is an essential in biological sequence design) ?

---

> ### Author Rebuttal · Authors · 2023-08-10
>
> We deeply appreciate the reviewer for the insightful and constructive comments!
>
> > How are the objectives computed? Is it “physics” based or does it rely on learning a surrogate model? Then, how are the generated molecules evaluated? Are the generated molecules evaluated on the objective using an actual oracle or learned surrogate function?
>
> For molecular optimization, the true objective values should be obtained by conducting expensive wet-lab experiments. However, real experimental evaluation makes the benchmarking of algorithms difficult and time-consuming. While physics-based simulation sounds rigorous, it consumes too many computing resources and may not be accurate enough to justify its extensive usage under many circumstances. Therefore, we follow the evaluation scheme adopted by prior related works (e.g., [1]) and use prediction models trained on an external dataset (not used for training HN-GFN) as our oracle functions.
>
> At each round of BO, during the training of HN-GFN, the generated molecules (trajectories) are evaluated using the learned surrogate function. Then the trained HN-GFN is used to propose a diverse batch of candidates, which are evaluated using the oracle.
>
> > What is the length of the generated molecules? Is it a hyperparameter of the model?
>
> We are sorry for our unclear description. The length is not a hyperparameter, we allow the agent to learn a special stop action and generate up to 8 fragments.
>
> > How sensitive are the results to the original dataset? In my personal experience, GFlowNets tend to be rather unstable or difficult to train due to the sparse feedback (and depends on the sequence size) and 200 initial molecules seem low to be able to learn a robust policy.
>
> We argue that the robustness of the policy will not be affected by the number of initial molecules for the following two reasons. 1) The 200 initial molecules are mainly used for training the surrogate models which determine the reward. 2) The training of GFlowNets primarily relies on trajectories obtained through online sampling. Moreover, in BO, the initial molecules are commonly randomly sampled, and their evaluation is also included in the budget. Hence, initial molecules are relatively low, different from offline model-based optimization where the performance is greatly influenced by the quality of initial molecules. For a fair comparison, we use the same fixed set of random initial molecules. To confirm our point, we conducted experiments with different sets of random initial molecules and found that the results are not sensitive.
>
> | | |GSK3β + JNK3 + QED + SA | |
> |:-|:-|-:|-:|
> |  | Starting HV | Final HV | Improvement |
> | HN-GFN (fixed)   | 0.101 | 0.416 ± 0.023 | 0.315 |
> | HN-GFN (random)  | 0.082 ± 0.018 | 0.402 ± 0.018 | 0.320 |
>
> > Does the proposed Hindsight-like off-policy strategy amounts to the proposition of [1], i.e. to hybrid the learning between generated molecules vs already generated molecules?
>
> Indeed, [1] only incorporates the trajectories from the available observed dataset, rather than those generated during training. Inspired by the empirical observation that GFlowNets benefit from offline trajectories, we adopt the concept of replay buffer which is widely used in RL to store and share high-performing molecules among policies by re-examining them with different preferences. Our strategy is delicately proposed for MOO, and has not been actively explored (in both RL and GFlowNet literature) prior to our work. Furthermore, the proposition of [1] is a special case of our strategy (when $\gamma=0$, the third paragraph in Sec 4.3). In Figure 2 (right), our strategy off-policy ($\gamma>0$) consistently outperforms [1] off-policy ($\gamma=0$).
>
> > What are the variables predicted by the MPNN?
>
> There are two MPNNs in our algorithm. 1) The MPNN serving as the policy network takes as input the partially constructed molecular graphs and predicts edge flow $F(s,s')$ and state flow $F(s)$. Specifically, for each stem of the graph (an atom where the policy can attach a new block), MPNN predicts $F(s,s')$ representing the unnormalized probability of attaching each block to this stem. For the stop action, we perform a global pooling and predict $F(s)$. 2) The MPNN serving as the surrogate model takes as input the complete molecular graphs and predicts multiple objectives simultaneously.
>
> > Can the authors discuss the choice regarding the flow network? For instance [1] use a MLP as a sampler.
>
> Following [2,3], we used MPNN for a fair comparison. In general, MPNN performs well in representing molecules. Other GNNs fit our framework as well. MLP is used for a fair comparison too in [1]. In my personal experience, 1D CNN and Transformer are better for sequence design.
>
> > A limitation of the paper is the lack of diversity in the experimental settings. [1] presents results for several benchmark datasets such as GFP. Can the authors comment on the ability of their method to adapt to such an offline setting (which is essential in biological sequence design)?
>
> Thanks for pointing this out. To our best knowledge, offline multi-objective benchmarks are rare. After reviewing related literature, we choose RFP from LaMBO [4] (optimize red-spectrum fluorescent protein sequences to maximize stability and SASA) and adapt the online setting to the offline setting. Following [1], we generate 128 candidates starting with the same $\mathcal{D}$ = 512 examples as [4]. Our HN-GFN significantly outperforms LaMBO.
>
> | |Relative HV|
> | :- | -: |
> |LaMBO|1.086 ± 0.010|
> |HN-GFN|**1.268 ± 0.089**|
>
> [1] Biological sequence designs with Gflownets.
>
> [2] Flow network based generative models for non-iterative diverse candidate generation.
>
> [3] Mars: Markov molecular sampling for multi-objective drug discovery.
>
> [4] Accelerating Bayesian optimization for biological sequence design with denoising autoencoders.
>
> **We hope our response can alleviate your concerns. Please let us know if you have any additional questions.**

---

> > ### Comment · Reviewer_9YCN · 2023-08-22
> > **Response to the authors**
> >
> > I thank the authors for their valuable response and additional technical details that significantly improve my understanding of their work.  I believe the discussion regarding the choice of the datasets could be included in the main paper.
> >
> > I raise my score to 5.

---

### Author Rebuttal · Authors · 2023-08-10

Dear Area Chairs and Reviewers,

We greatly appreciate the reviewers' time, valuable comments, and constructive suggestions. Overall, the reviewers deem our paper as "well-organized" (9YCN, BkDd, hutP), studying "an important problem" (9YCN, BkDd, Dc63), acknowledging our methodology novelty and soundness (9YCN, BkDd, hutP, Dc63), experiments with strong experimental results (hutP, Dc63) and ablation study (hutP, Dc63), demonstrating the potential use and advantages of the proposed method (9YCN, BkDd).

In the author response period, we made diligent efforts to address reviewers' concerns and provided additional experimental results to further verify our contributions. The summary of our main efforts is presented as follows:

- We have provided a detailed explanation for the protocol, experimental details, and implementation details. (To 9YCN, BkDd, hutP, and Dc63)

- We have further conducted diverse experiments (offline biological sequence design and different molecular optimization benchmarks) to demonstrate the general applicability of the proposed method (To 9YCN and BkDd)

- We have further conducted experiments to analyze the impact of the initial dataset to demonstrate that the results are not sensitive to the initial dataset (To 9YCN)

- We have further discussed and conducted experiments to analyze the impact of batch size to demonstrate that our setting (a batch size of 100 and 8 rounds) is reasonable and suitable for the design and benchmarking of algorithms for molecular optimization (To BKDd and Dc63)

In our individual responses, we provide detailed answers to all the specific questions raised by the reviewers. Further discussions are welcomed to facilitate the reviewing process toward a comprehensive evaluation of our work.

---

### Decision · Program_Chairs · 2023-09-21

**Decision:**

Accept (poster)

**Comment:**

This paper proposes a new GFlowNet framework for multi-objective optimization of molecules. The reviewers appreciate the importance of the problem, presentation, and the idea of using GFlowNets (which can produce diverse outputs) for multi-objective optimization.

After reading the paper myself, I particularly enjoyed the idea of using hyper-networks to condition GFlowNets on the preference vectors. In terms of multi-objective molecular optimization, I think the empirical evaluation could be improved by adding more competitive baselines, e.g., Winter et al., 2019. However, despite the moderate empirical evaluation, I believe this work provides a useful contribution to the GFlowNet community, so I would recommend acceptance for this work.

Finally, as the authors mentioned, there is a concurrent work that already appeared at ICML 2023 that shares the idea of using GFlowNets for multi-objective optimization. There also exists the Pareto GFlowNet that shares a similar idea with the paper. While I agree that this work provides new contributions even when compared with the ICML 2023 paper and the Pareto GFlowNet, I recommend the authors add a thorough discussion with respect to these papers, once the paper is accepted. An empirical comparison would be even more significant and beneficial for the community.

[Winter et al., 2019] Efficient multi-objective molecular optimization in a continuous latent space